# THE SAME BUT DIFFERENT: STRUCTURAL SIMILARITIES AND DIFFERENCES IN MULTILINGUAL LANGUAGE MODELING

**Ruochen Zhang**[1][*]  **Qinan Yu**[2][*][†]  **Matianyu Zang**[1]  **Carsten Eickhoff**[3]  **Ellie Pavlick**[1]

[1]Brown University  [2]Stanford University  [3]University of Tübingen

## ABSTRACT

We employ new tools from mechanistic interpretability to ask whether the internal structure of large language models (LLMs) shows correspondence to the linguistic structures which underlie the languages on which they are trained. In particular, we ask (1) when two languages employ the same morphosyntactic processes, do LLMs handle them using shared internal circuitry? and (2) when two languages require different morphosyntactic processes, do LLMs handle them using different internal circuitry? In a focused case study on English and Chinese multilingual and monolingual models, we analyze the internal circuitry involved in two tasks. We find evidence that models employ the same circuit to handle the same syntactic process independently of the language in which it occurs, and that this is the case even for monolingual models trained completely independently. Moreover, we show that multilingual models employ language-specific components (attention heads and feed-forward networks) when needed to handle linguistic processes (e.g., morphological marking) that only exist in some languages. Together, our results are revealing about how LLMs trade off between exploiting common structures and preserving linguistic differences when tasked with modeling multiple languages simultaneously, opening the door for future work in this direction.

## 1 INTRODUCTION

As large language models (LLMs) have become the undisputed state of the art for building English language technology, there is decided interest in replicating their success across the full range of human languages. However, very little is known about the internal structure of LLMs, and whether such structure is conducive to acquiring broad multilingual capabilities. In fact, recent research has produced seemingly contradictory findings, such as evidence that multilingual models adopt language-specific representations (Tang et al., 2024; Choenni et al., 2024), while simultaneously showing good transfer across languages even in cases that would appear to have no superficial similarities that can be exploited to aid such transfer (Pires et al., 2019). Given the importance of building technology for diverse languages, there is a need for a more precise understanding of how LLMs represent structural similarities and differences across languages, and whether such representations accord with our intuitive understanding of how languages work.

In this paper, we present a focused case study on the relationship between English and Chinese processing in LLMs. We employ tools from the growing subfield of *mechanistic interpretability* in order to ask whether the internal structure of LLMs show correspondence to the linguistic structures which underlie the languages on which they are trained. We focus on only the most minimal criteria of correspondence. In particular, we ask (1) when these two languages employ the same morphosyntactic processes, do LLMs handle them using shared internal circuitry? and (2) when these two languages require different morphosyntactic processes, do LLMs handle them using different internal circuitry? While these questions seem simple, their answers are non-obvious. LLMs readily employ overlapping circuitry for tasks that do not necessarily seem "the same" to humans

---

[*]Equal Contribution. Correspondence to ruochen_zhang@brown.edu.
[†]Work done at Brown University.

(Merullo et al., 2024), and at the same time, neural networks frequently differentiate concepts due to surface form variation (Olah et al., 2020), even when humans would easily identify them as part of the same abstract category.

Using English and Chinese multilingual and monolingual models, we analyze the internal circuitry involved in two tasks, one focusing on indirect object identification (IOI) which is virtually identical between the languages, and one which involves generating paste tense verbs that require morphological marking in English but not in Chinese. Our contributions are as follows:

- We show that a multilingual model uses a single circuit to handle the same syntactic process independently of the language in which it occurs (§3.4).
- We show that even monolingual models trained independently on English and Chinese each adopt nearly the same circuit for this task (§3.5), suggesting a surprising amount of consistency with how LLMs learn to handle this particular aspect of language modeling.
- Finally, we show that, when faced with similar tasks that require language-specific morphological processes, multilingual models still invoke a largely overlapping circuit, but employ language-specific components as needed. Specifically, in our task, we find that the model uses a circuit that consists primarily of attention heads to perform most of the task, but employs the feed-forward networks in English only to perform morphological marking that is necessary in English but not in Chinese (§4).

Although the scope of our study is restricted to one language pair and a few phenomena, our results provide informative insights into how LLMs trade off between exploiting common structures and preserving linguistic differences when tasked with modeling multiple languages simultaneously. Our experiments can lay the groundwork for future works which seek to improve cross-lingual transfer through more principled parameter updates (Wu et al., 2024), as well as work which seeks to use LLMs in order to improve the study of linguistic and grammatical structure for its own sake (Lakretz et al., 2021; Misra & Kim, 2024).

## 2 ANALYSIS METHODS

In this work, we are interested in analyzing how large language models (LLMs) trained in different languages differ in terms of the algorithms and mechanisms they invoke to handle various aspects of language processing. To do this, we employ a few recently developed analysis techniques, described below. These techniques are similar in spirit, but differ in certain details that matter for our analysis. For the most part, we find converging evidence for the paper's main claims across both techniques. When results differ in interesting ways, we comment in our results sections.

### 2.1 PATH PATCHING

Path patching (Wang et al., 2023; Goldowsky-Dill et al., 2023; Vig et al., 2020; Hanna et al., 2023; Tigges et al., 2023b) has become the most standard and widely-accepted technique within the still-new subfield of *mechanistic interpretability*. The goal of path patching is to localize specific *circuits* within the weights in a trained neural network that play a causal role in model behavior. The setup requires a pair of contrastive inputs, one referred to as the *clean* input and the other as the *corrupted* input. Path patching caches the activations for both inputs and then replaces the values of individual heads on the clean input with the values those heads would have taken had they been run on the corrupted input. In this way, the method aims to find the specific important head which maximally explains the final logits. Working backward, i.e., through patching the important heads at each layer, path patching has been used to identify full circuits that carry out the task. On its own, path patching only identifies important heads. To gain insight into the specific functions of these heads, path patching is usually used with logit attribution (Nostalgebraist, 2020; Belrose et al., 2023; Dar et al., 2023; Yu et al., 2023) which projects activations into the vocabulary space, as well as with bespoke analysis techniques invented by prior work to explain specific types of heads, such as duplicate-token detection heads or copy heads (Wang et al., 2023).

The advantage of path patching is primarily its wide adoption, which makes it easier to trust results, and enables us to compare with prior work in order to vet the results we are seeing (i.e., checking that we reproduce prior work when we expect to do so). The primary downside is that the method

requires careful design of the minimal pair templates. The circuit that is discovered is highly dependent on how these pairs are defined. Moreover, in some of our language pairs, a comparable design of such templates is not possible (see § 4). Thus, we explore additional techniques.

## 2.2 INFORMATION FLOW ROUTES

Given the aforementioned drawbacks of path patching, we also employ the information flow routes method (Ferrando & Voita, 2024; Tufanov et al., 2024). At each timestep, this method computes the contribution of each head to the residual streams. These heads are then aggregated over the entire attention block to construct a graph of the information flow that show which residual stream at which layer is important to the current residual stream value update across different timesteps. In this method, different from path patching where we can see circuit components across layers, the importance can only be computed every two layers. Compared to path patching, information flow routes have the advantage of not requiring minimal pairs of inputs. The tradeoff is that (1) the method is new and thus we do not benefit from clear expectations about what it should yield in certain settings and (2) this method tends to discover more generic components and larger circuits. In our paper, we thus use both information flow routes, and path patching along with logit attribution in conjunction for analysis at different granularity levels. Unless otherwise stated, both methods produce results that are consistent with the primary claims we make in each experimental section.

## 3 TASKS WITH COMMON LINGUISTIC STRUCTURE ACROSS LANGUAGES

### 3.1 QUESTIONS

We first ask whether large language models (LLMs) will learn to use shared circuitry for different languages, specifically for aspects of language processing that reflect similar structures across languages. That is: if two distinct languages use similar morphosyntactic structure but realize it using different lexical items, will an LLM treat these as the same (handling them with the same abstract internal circuit) or as different (handling them with distinct, language-specific circuits)? We ask this question first in the case of multilingual models, and second in the case of monolingual models.

### 3.2 INDIRECT OBJECT IDENTIFICATION (IOI) TASK

We set up our data following the original IOI paper (Wang et al., 2023). IOI consists of sentences such as *"Susan and Mary went to the bar. Susan gave a drink to [BLANK]"*, in which the model is expected to predict *Mary*. These names are tokenized as one single token in the models. To diversify the data, we use fifteen different templates for the actions. We also use ChatGPT to translate the templates and names into Chinese and have them manually inspected by native speakers.

Wang et al. (2023) identifies a specific circuit within GPT2 (Radford et al., 2019) for performing the IOI task. At a high level, the circuit reveals that the model runs the following algorithm: first, it identifies that there is a duplicated name (in the above example, *Susan*). Second, it inhibits attention to this duplicated name. Third, it copies the remaining (non-duplicated) name. These steps are carried out by a set of functionally specialized attention heads: Previous Token Heads, Duplicate Token Heads, Induction Heads, S-inhibition Heads, and Name Mover Heads. Duplicate Token Heads attend to the token position of *Susan* and identify this name token is mentioned twice in the sentence. S-inhibition Heads inhibit the Name Movers' attention to both occurrences of *Susan*. The Name Mover Heads output the remaining name (*Mary*). More detailed descriptions of these heads are given in prior work (Wang et al., 2023; Merullo et al., 2024). For our purposes, what is important is whether the same heads are important and perform the same roles across languages.

### 3.3 MODELS

We use BLOOM-560M[1] (Workshop et al., 2022) as our multilingual model. We first evaluate whether BLOOM is able to complete the task by checking if it always prefers the correct name over the incorrect name. We use both accuracy and zero-rank rate to measure the model performances. Accuracy computes the percentage of times when the probability of predicting the correct

---

[1]We refer to the model as BLOOM for simplicity below.

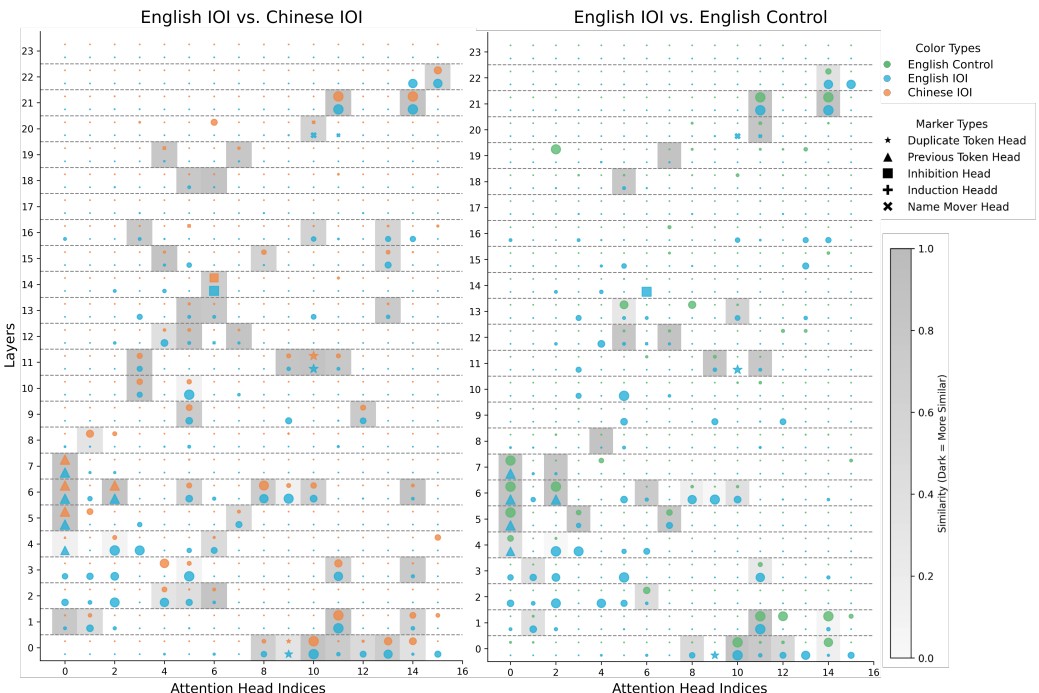

Figure 1: Attention heads activation frequency comparison between English IOI, Chinese IOI and English control tasks. Left: Comparison between English IOI and Chinese IOI on BLOOM (with specific functional heads in different marker types). Right: Comparison between English IOI and English Tense on BLOOM. For pairs of heads with non-zero activation frequency, they are shaded based on their value difference. Darker gray means smaller differences. The left graph has more number of shaded head pairs compared to the right, indicating a greater similarity between the activated heads.

name is higher than the other name. The zero-rank rate measures the percentage of times where the correct name is the actual top answer. In the IOI task, BLOOM has an accuracy of 100% and a zero-rank accuracy of 96% in English and 95% accuracy and 93% zero-rank accuracy in Chinese. As a multilingual model, BLOOM has a high and comparable performance between Chinese and English.

For experiments with monolingual models, we select GPT2-small (Radford et al., 2019) as our English-only model and CPM-distilled [2] as our Chinese-only model[3]. These two models have the same architecture and contain the same number of parameters. GPT2-small has a 99.5% accuracy and CPM has an 84.5% accuracy. The zero-rank rate for GPT2 is 97.5% meaning that the correct name has the highest probability most of the time. CPM has a zero-rank accuracy of 57.5%. Both models are able to complete the task while the performance of GPT2 is comparably better.

### 3.4 RESULTS ON MULTILINGUAL MODEL

We first validate whether the English and Chinese tasks share the same important heads. We use the information flow routes technique across 50 examples respectively in English and Chinese. We calculate the activation frequency for every head component. If one head is highly activated[4] then it indicates this head is important for the task. In Figure 1, we observe a high overlap in the important heads used between languages. The heads that are shown to be important to solve the task in English are also important in Chinese. As a comparison, we compare with the BLOOM's activated heads

---

[2]https://huggingface.co/mymusise/CPM-Generate-distill

[3]We refer GPT2-small as GPT2 and CPM for CPM-distilled below for simplicity.

[4]Following Ferrando & Voita (2024), we set the threshold $\tau = 0.03$. We consider a head is highly activated if its contribution value surpasses $\tau$.

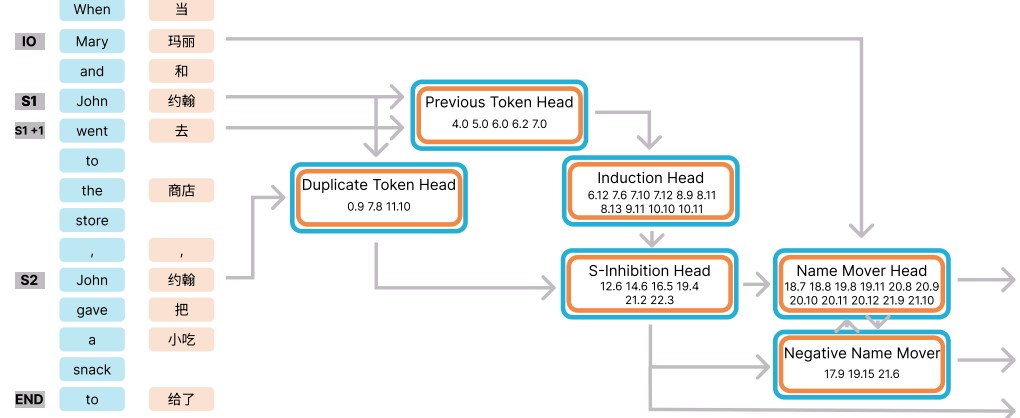

Figure 2: Illustration of IOI circuits of English and Chinese on BLOOM-560M model. Blue-framed rectangles or texts are the components used in English IOI and orange-framed ones are for Chinese IOI. Heads marked in black are the shared functional heads between English and Chinese IOI. The English and Chinese IOI circuits are highly similar as they use most of the same components for the same functionality to implement the algorithms.

from an unrelated task (specifically, the past tense task described in § 4.2) as a control. If the circuit overlap is indeed indicative of the model exploiting shared algorithmic structure, we would expect that the overlap between English and Chinese variants of the same linguistic task should be higher than the overlap between English for two tasks that invoke dissimilar linguistic (i.e., syntactic and semantic) structures. Indeed, compared with the high overlap between the IOI task in Chinese and English, significantly different heads are used in the control task. To measure the similarity between the activation patterns, we compute the Pearson Correlation Coefficient $\rho$ (Freedman et al., 2007) between the head activation frequency for the two task pairs. While $\rho$ between English and Chinese IOI task is 0.72, $\rho$ between English IOI and English control task is 0.48. Anecdotally, we observe that many of the heads that are shared between English IOI and control are also shared with Chinese IOI. This suggests that these might be generically important heads which we expect to appear in a wide range of tasks, although further work is needed to test this hypothesis quantitatively.

We continue to examine if these shared functionalities are the same as the ones defined in Wang et al. (2023). In Figure 2, we identified functionalities of the important components in Figure 1 in Chinese and English in BLOOM. We first use path patching to identify the components and then use the specific metrics mentioned in Wang et al. (2023) to validate their functionalities. Both of these experiments return highly similar results[5]. The functionalities we identify align with Wang et al. (2023). This suggests that not only is the mechanism (i.e., the heads and their functionality) shared across English models (GPT2 and BLOOM), but moreover this same mechanism is shared nearly exactly between Chinese and English within BLOOM.

## 3.5 RESULTS ON MONOLINGUAL MODELS

While the cross-lingual circuit overlap in a multilingual model is not obvious (i.e., it requires abstracting over patterns that manifest using disjoint sets of surface forms), it is also reasonable to assume that the convergence stems from the model's broader multilingual training data and objective. We thus ask whether similar patterns arise even when models are trained independently in English and in Chinese. For this analysis, as mentioned, we use the English-only model GPT2-small and a distilled version of the Chinese-only model CPM-Generate (Zhang et al., 2020) for their exact same architecture.

We discover circuits with path patching which gives similar results and identifies the same components that are important for the circuits. In both GPT2 and CPM, we identified similar circuits as Wang et al. (2023) outlined in Figure 3. This is surprising as it suggests that at a high level, the

---

[5]Detailed analysis can be found in §B.1

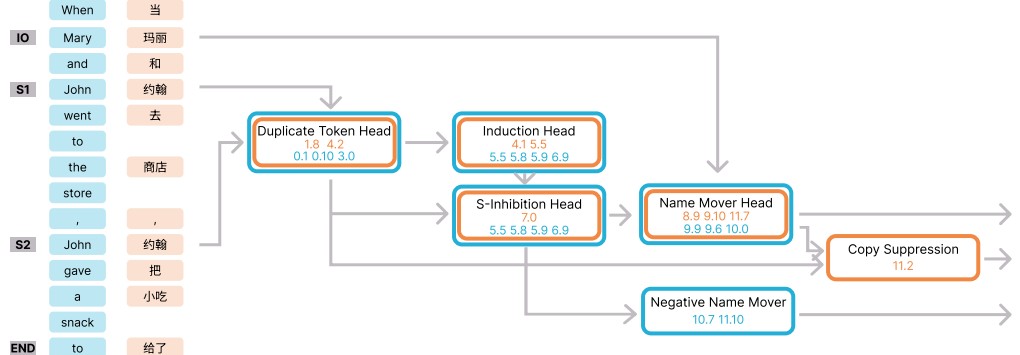

Figure 3: English and Chinese IOI task circuits. Blue highlights are components used and inputs for the English task in GPT2. Orange highlights are for CPM. The components with both blue and orange frames are shared in both the English and Chinese circuits. In each of these heads, blue texts indicate the heads in the English Circuit and orange in Chinese. The Negative Name Mover heads appear only in the English model. Copy Suppression Head appears only in the Chinese model. Despite trained on completely different data, models still implement largely similar algorithms to solve tasks in different languages.

LLMs consistently converge to highly similar algorithms for solving this type of task, even when trained on very different languages. That said, we do observe some differences between the circuits. In GPT2, Wang et al. (2023) identifies Negative Name Mover Heads. These heads write negatively in the direction of the name tokens. Similarly, Negative Name Mover Heads are found in the circuit in GPT2. However, we didn't observe any Negative Name Mover Heads in CPM, instead of observed the Copy Suppression Head. Such heads are identified by McDougall et al. (2023b) in the Pythia model (Biderman et al., 2023). Instead of reading the information flow from the S-inhibition head, Copy Suppression Heads read directly from the Duplicate Token Heads and suppress the logit of the repeated token in the output. This divergence could provide a good starting point for future work on predicting when and how specific circuitry emerges during training (see Discussion §5).

## 4 TASKS WITH LANGUAGE-SPECIFIC STRUCTURE

### 4.1 QUESTIONS

The above results show that LLMs are capable of recognizing structural parallels across languages, and even that monolingual models, trained without explicit influence from other languages, converge on similar algorithms for similar structures. We next ask: what happens when languages exhibit dissimilar structures? In reality, languages exhibit immense morphological and syntactic variation, and there is never a perfect one-to-one correspondence between the structural elements of different languages. A straightforward example of this phenomenon is the fact that English includes morphological markers for tense (e.g., *walk* vs. *walked*) while Chinese does not. We thus ask, given such variation, do LLMs adopt different circuitry for different languages, or rather do they continue to invoke shared circuitry but employ language-specific subroutines as needed?

### 4.2 PAST TENSE TASK

We construct minimum pair templates in English: "*Now I verb$_{present}$. Yesterday I also verb$_{past}$.*" and "*Yesterday I verb$_{past}$. Now I also verb$_{present}$.*". For Chinese, we construct the exact same template in Chinese "现在我 *verb$_{present}$*。昨天我也 *verb$_{past}$*。" and "昨天我 *verb$_{past}$*。现在我也 *verb$_{present}$*。"[6]. To collect the verbs, we start with the English verb tense dataset from Big-bench (Srivastava et al., 2022) and filter the verbs to only retain those with regular inflection rules. Then all English verbs are translated, deduplicated and filtered with native speakers' manual inspection. Notice that in Chinese, there are no morphological markers between *verb$_{present}$* and *verb$_{past}$*. Both of them refer to the stem

---

[6]现在-Now，昨天-Yesterday, 我-I, 也-also

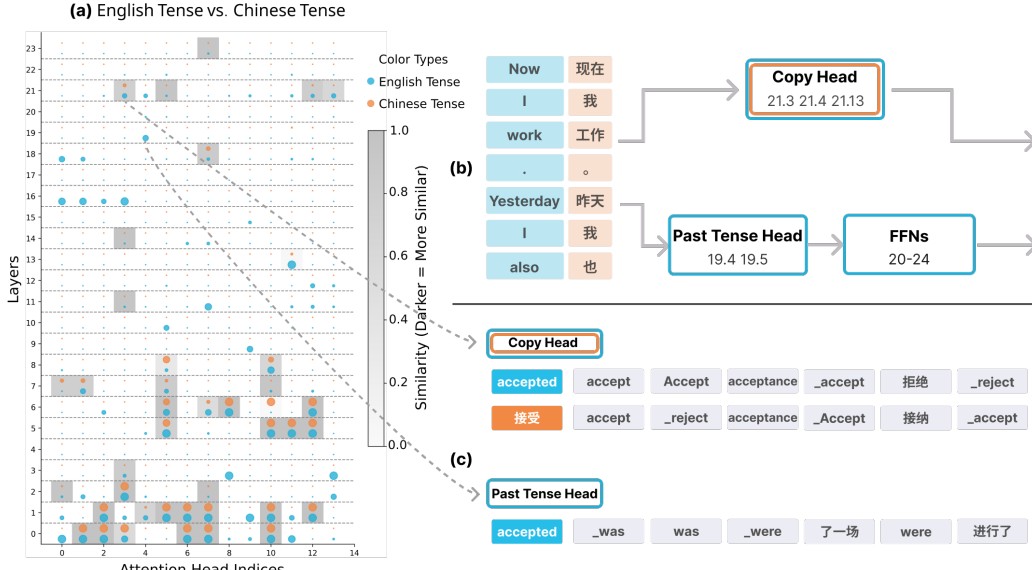

Figure 4: (a) Head activation frequency comparison between English and Chinese past tense task on Qwen. (b) Illustration of the tense circuits in Qwen that contains three components we focus on: copy heads, past tense heads and feed-forward layers. (c) Top promoted tokens from copy head and past tense head. Words highlighted in either blue and orange are expected model predictions. Gray tokens are the specific tokens that the head promotes. Here the copy head is 21.3 and the past tense head is 19.4. As shown in (a), the copy head has a similarly high activation frequency in both languages whereas the past tense head is only frequently activated in English.

form of the verb. By leveraging this distinction between the English and Chinese languages, we are able to investigate the mechanistic differences required when models process semantic equivalent tasks but with different morphosyntactic rules. Our analysis on this task focuses more specifically on multilingual models instead of monolingual models since they allow us to more directly observe the structral similarities between the circuitries.

### 4.3 MODELS

We use Qwen2-0.5B-instruct[7] (Yang et al., 2024) for the task for its better benchmark performance compared to BLOOM[8]. The final verb collection contains 62 English verb instances and their Chinese translation and themselves are both single tokens after being processed by the Qwen tokenizer. For the English task, Qwen reaches 96.77% accuracy for the normal input and 58.06% for the zero-rank rate. In Chinese, as there are no minimal pairs, we can only obtain a zero-rank rate of 25.81%.

### 4.4 RESULTS

In Figure 4, compared to the active heads distribution on the IOI task, we notice that some early-layer heads have high activation frequencies that are shared in both English and Chinese tasks. However, there are much fewer shared heads in the later layers of the model. Specifically for the Chinese task, we observe that almost no heads have high activation frequency in the later layers (Layer 19-20). We posit that some heads that are shown to be more active in these later layers are responsible for the additional inflection rule unique to the English task in our setting, which is unnecessary to the Chinese task.

---

[7] We refer to the model as Qwen below for simplicity.

[8] Here we replace the model of interest from BLOOM to Qwen. Qwen is similarly a multilingual language model but with a strong language focus on both English and Chinese. It achieves comparable performances in English compared to BLOOM but significantly better performance in Chinese, which is necessary for extracting the circuits. See Table 1 in the Appendix for the performance comparison.

## 4.5 FUNCTION-SPECIFIC HEADS

Through path patching, we observe that heads 19.4 and 19.5[9] exert the most significant influence on the final logits (Refer to Appendix C for details). Examining the top promoted words from the most positive projections into the model vocabulary space (Figure 4 (c)), we find that they generally favor past-tense words in English (e.g., was, were, and other past-tense verbs unrelated to the target verb). Among the top-promoted tokens, we also notice Chinese verbs with the suffix "了" (e.g., 进行了，can be translated to "done".), where "了" is commonly used to indicate the completion of an action. Even when provided with Chinese input, these heads continue to strongly encode past-tense concepts. In Figure 4(a), head 19.4 appears frequently activated for English but not for Chinese, suggesting that these heads are predominantly engaged in English processing rather than in Chinese.

To validate our findings, we ablate[10] the heads 19.4 and 19.5 and analyze their impact on the tasks. The rank of correct past tense verbs in English remains relatively unchanged. Whereas other non-relevant past tense verbs move backward slightly to make spaces for the corresponding present tense verbs, resulting in their ranks getting promoted by an average of 83.21 positions. In the Chinese task, the rank of the correct verb moves forward by an average of 4.58 positions. Notably, after ablating these past-tense heads and projecting the final-layer logits to the vocabulary space, present-tense verbs emerge as the second most probable token (See Figure 12 for details). These results suggest that in English, past-tense heads actively suppress present-tense verbs to disambiguate between verb tenses. However, these heads do not play a similar role in the Chinese task.

In Ferrando & Voita (2024), they mention that patching can only emphasize the heads that are important for the original tasks but not the contrastive baseline. In the past tense task, we locate past tense heads that are in charge of disambiguating past tense verbs from present tense ones via path patching. These heads are discovered due to the additional morphological markers between the original and contrastive sentence. However, how do models know to promote that specific verb (with or without the marker) without confusing it with others?

To address this question, we compute the total score of each head projecting onto either past tense or present tense verb groups (see Appendix C.1). The primary heads identified are 21.3, 21.4, and 21.13. We observe that these heads not only emphasize different morphological variants of the target verb but also promote tokens that are orthographically similar or semantically related, such as synonyms and antonyms (see Figure 12 in Appendix). These heads are crucial for both English and Chinese tasks.

To assess their impact, we perform ablation studies on these heads and observe significant impacts on model performance. In English, the average rank of past tense verbs drops by 141.95, while present tense verbs experience a much larger average demotion of 221.39. This disparity suggests that past tense verbs are less affected, possibly because the identified past tense heads remain active, suppressing present tense verbs. As shown in Figure 12, the top-promoted words after ablation are still predominantly general past tense verbs. For Chinese, the model performance also declines, with verbs in general experiencing an average rank demotion of 201.82. These ablation results indicate that the copy heads are essential for both English and Chinese tasks, while the past tense heads are specifically activated in English, where they are crucial for task completion. However, qualitative inspection also suggests that the semantics of the verb is not entirely decoupled from the tense, highlighting a need for further work in order to produce more nuanced descriptions of the function of these heads in this circuit, and in general.

## 4.6 FEED-FORWARD NETWORKS

Previous work (Merullo et al., 2023) discusses the role of the feed-forward networks (FFN) in retrieving the correct transformation required for the past tense tasks. Specifically, these FFNs are found in the last few layers of the models where the ranked position of the present-tense verb and past-tense verb switch. Comparing the English and Chinese tasks, we assume that these FFN layers play a role in predicting the past-tense verb in English but remain unactivated for the Chinese task. When we ablate out the FFNs from layer 20-24, the accuracy on the English task drops from

---

[9]We use layer.head for referring to specific heads of interest.

[10]The ablation experiments carried out in the paper are done via zero-ablation, which refers to zeroing out the contribution of a certain head.

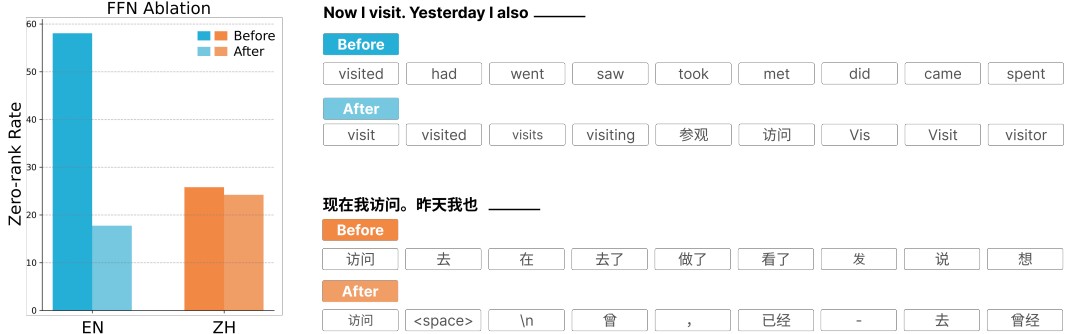

Figure 5: Illustration of the effects of late feed-forward layers ablation (layer 20-23). Left: Zero-rank rate changes comparison between English and Chinese past tense task. Right: Top predicted tokens comparison for before and after ablation. We observe that English performance is significantly impacted after ablation but Chinese is barely influenced. Qualitative results on the right show that the correction token move backward in English but remain the top prediction in Chinese.

97.44% to 47.44% and the zero-rank rate from 58.06% to 17.74%. However, for the Chinese task, the zero-rank rate is barely changed (25.08% to 24.19%).[11]

In the English example, when examining top predicted tokens qualitatively in Figure 5, we see that the rank of the correct past tense answer *visited* moved back by 1 position after ablating the MLP layers. Additionally, it is worth noting that morphological variants (E.g. *visit*, *visits*, etc.) or semantically relevant tokens (E.g. 参观, 访问[12]) move to the top predictions. These are also tokens that are actively promoted by the copy heads mentioned earlier. In the Chinese example, we see that the top token as the correct prediction still remain at the same position, unaffected by the FFN layers removal.

## 5 RELATED WORK AND DISCUSSION

**Mechanistic Interpretability**    We built on previous work on mechanistic interpretability to reverse engineer the mechanism of neural network. *Circuits* are a significant paradigm of model analysis that has emerged from this field. Originated from the vision model (Olah et al., 2020) and continued to transformer language model (Meng et al., 2023; Wang et al., 2023; Hanna et al., 2023; Varma et al., 2023; Merullo et al., 2024; Lieberum et al., 2023; Tigges et al., 2023a), works in mechanistic interpretability try to characterize the function of individual components in completing certain tasks. Beyond the work in understanding the role of attention heads (Olsson et al., 2022; Chen et al., 2024; Singh et al., 2024; Gould et al., 2023; McDougall et al., 2023a), efforts are made in understanding neurons and feed-forward network in (Vig et al., 2020; Finlayson et al., 2021; Sajjad et al., 2022; Gurnee et al., 2023; Voita et al., 2023; Merullo et al., 2023). These studies mainly use causal methods with minimal pairs to locate the components and their functionalities (Vig et al., 2020; Chan et al., 2022; Geiger et al., 2021; 2023; Meng et al., 2023; Wang et al., 2023; Chan et al., 2023; Cohen et al., 2023). However, these methods heavily rely on manual inspection, which limits their scalability to larger models. Therefore other recent works have attempted to automate the causal process (Conmy et al., 2023; Bills et al., 2023; Syed et al., 2023; Hanna et al., 2024) or use input attribution to trace the subgraph of important components lifting the burden of defining minimal pairs (Ferrando & Voita, 2024).

**Multilingual Interpretablity**    With English-centric large language models demonstrating outstanding generation and reasoning capabilities, more current efforts have been devoted to expanding such capabilities to multilingual settings (Workshop et al., 2022; Yang et al., 2024; Üstün et al., 2024; Aryabumi et al., 2024). Many works (Wu & Dredze, 2019; Winata et al., 2022) have lever-

---

[11]Note we cannot compute accuracy in Chinese as there is no past-tense alternative verb form against which to compare.

[12]Both words can be translated to *visit*.

aged transfer learning to boost performances on low-resource languages as an alternative to tackling data deficiency problems. Meanwhile, language models have also been reported to suffer from the "curse of multilinguality" (Conneau, 2019; Chang et al., 2023) where languages interfere with each other given fixed model capacities.

Previous works have attempted to investigate cross-lingual representation alignment in BERT-based models (Artetxe et al., 2020; Muller et al., 2021; Conneau et al., 2020; Del & Fishel, 2021; 2022; Hämmerl et al., 2024), observing the "First Align, the Predict" paradigm (i.e. models first converge cross-lingual representation to language neutrality then become language specific again across the layers). Other works also quantify their effects on transfer learning abilities on downstream tasks (Malkin et al., 2022; Choenni et al., 2023). More recent works such as Wendler et al. (2024) observe that multilingual generative models go through language conversion phrases with the reasoning process mainly conducted in English. Tang et al. (2024) specifically locates these language-specific neurons and manipulates the output language by activating or deactivating different language-specific neurons. Li et al. (2024) also observe that Multi-Layer Perceptron (MLP) is associated with how toxic concepts are encoded crosslingually. However, it remains unclear how models utilize their components when receiving inputs in different languages: when a task is prompted in different languages, are the underlying circuits shared or does there exist completely separate paths? While Merullo et al. (2024) has reported the reuse of circuit components across different tasks in the same language, no prior work has investigated this phenomenon at the cross-lingual level. Additionally, considering language construction is unlikely to be exactly the same across languages, how are model components used to reflect these differences? In this paper, with the help of mechanistic interpretability techniques like path patching and information flow routes, we provide an initial analysis to quantify to what extent model components are shared when solving tasks for different languages.

We investigate when and how large language models (LLMs) encode structural similarities between languages. Specifically, we ask (1) whether linguistic tasks that are structurally similar across languages are handled by the same LLM circuitry and (2) whether linguistic tasks that employ language-specific structure are handled by specialized LLM circuitry. Through a series of experiments employing different tasks and models in both English and Chinese, we find that indeed cross-lingual structural similarity is represented within the LLM using shared circuits which invoke the same algorithm independent of language. We find that these shared algorithms emerge even in monolingual models trained on different languages. We also show that when tasks require language-specific components–specifically past tense morphology in English–LLMs localize the corresponding processing such that it plays a causal role in some languages but not in others.

This work builds on previous literature (Artetxe et al., 2020; Muller et al., 2021; Conneau et al., 2020) that discuss the "First Align, then Predict" paradigm, where parameter sharing between models allows for better alignment of cross-lingual structures. Our work in generative models provide supportive evidence by diving deeper into the specific attention and FFN weights that are responsible for such structural similarities. Taken together, our results provide a first step in using mechanistic analysis about LLMs' inner workings in order to ask higher-level questions about how linguistic structures are reflected in state-of-the-art AI models.

Our results are limited to specific tasks and datasets, future work will be necessary to replicate and generalize our findings. Even so, our studies yield several interesting and encouraging insights which can inspire new lines of inquiry. For example, we find that the same circuit emerges across different models trained on entirely different languages. This raises questions about what type of structure is shared across otherwise-dissimilar languages, or even shared universally? That is, what regularities exist in the distribution of both English and Chinese such that they give rise to the same highly specific functional components organized in the same way? While the details of this discussion are specific to LLMs, the spirit shares much with long-running linguistic discussions of universality and of learnability in language (Yang, 2004). Deeper cross-disciplinary collaborations could possibly exploit LLMs in order to gain fresh perspectives on such questions. Another important direction concerns practical considerations for developing state-of-the-art technology in languages other than English. This includes not only building LLMs in other languages (Wu et al., 2024), but also ensuring their safety and robustness (Yong et al., 2023; Deng et al., 2023) across languages. Insights into specific mechanisms that are shared between languages could guide more principled approaches to transferring parameters or anticipating failure modes. Future work could explore whether circuit overlap like what we document here can be exploited towards these goals.

## 6 ACKNOWLEDGEMENTS

We thank Benjamin Muller, Zhaofeng Wu, Javier Ferrando, Yong Zheng-Xin, Jacob Li, Rui-Jie Yew, members of the Lunar lab and Health NLP lab at Brown, and the anonymous reviewers for their valuable discussions and feedback on this work.

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

## A  TASK BENCHMARK PERFORMANCES

In this section, we present the accuracy and zero-rank rate of both the IOI tasks and Past Tense tasks discussed in § 3 and § 4. In Table 1, we see that all monolingual models and multilingual models perform well on the IOI tasks. For the past tense task, BLOOM obtains good performance in English but not in Chinese. In comparison, Qwen performs slightly worse in English but much better in Chinese. As a result, we focus on Qwen for our language-specific analysis.

| Model | Lang. | Accuracy | | Zero-Rank Rate | |
|---|---|---|---|---|---|
| *IOI Task* | | | | | |
| | | normal | flipped | normal | flipped |
| GPT2-small | EN | 99.5 | 99.5 | 97.5 | 95.5 |
| CPM-Generate-distill | ZH | 84.5 | 86.5 | 57.5 | 73.0 |
| BLOOM-560M | EN | 100.0 | 100.0 | 96.0 | 91.0 |
| BLOOM-560M | ZH | 95.0 | 100.0 | 93.0 | 99.5 |
| *Past Tense Task* | | | | | |
| | | normal | flipped | normal | flipped |
| BLOOM-560M | EN | 90.3 | 51.6 | 80.7 | 37.1 |
| BLOOM-560M | ZH | - | - | 12.9 | 1.6 |
| Qwen2-0.5B-instruct | EN | 96.8 | 69.4 | 58.1 | 24.2 |
| Qwen2-0.5B-instruct | ZH | - | - | 25.9 | 21.0 |

Table 1: Model performances for the IOI and past tense tasks. Note that we are unable to compute accuracy for the past tense task in Chinese (marked by -) due to the absence of minimal pairs.

## B  PATCHING DETAILS OF THE IOI TASK

### B.1  MULTILINGUAL MODELS

To extract the IOI circuits on the BLOOM model, we use the transformer-lens library (Nanda & Bloom, 2022) and follow the practices of Wang et al. (2023). Figure 6 shows the main steps to discover specific function heads for both English and Chinese tasks on BLOOM. Throughout each of the steps, we notice the important heads discovered between English and Chinese are highly similar, leading to largely overlapping circuits. Figure 7 validates the functions of the Duplicate Token Heads, Previous Token Heads and Induction Heads found by showing their attention scores on sequences of random tokens. Please refer to more detailed definitions and experiment designs on these metrics in Wang et al. (2023). Additionally, we provide IOI circuits identified in Qwen2-0.5B-instruct by carrying out the same procedure as above (shown in Figure 8).

### B.2  MONOLINGUAL MODELS

For path patching on CPM, we add the CPM models to the transformer-lens library and follow the same steps as the multilingual model described above. Figure 9 presents the comparison between English and Chinese inputs when we patch to the residual stream on GPT2 and CPM respectively. Notice that here the heatmap patterns are quite different because they are trained on completely different data despite the same architecture (with the same number of layers and heads). Then we perform patching to individual heads and follow Wang et al. (2023) to find out their functionalities. Figure 9 Right shows the attention patterns of the name mover heads and copy suppression heads. We see that mover heads `8.9` and `9.10` show copying behaviors toward the IO tokens and `11.7` shows copying behavior toward both IO and S tokens. The copy suppression head `11.2` on the other hand suppresses the copying action of the S token. Figure 10 shows our validation experiments on the functions of Induction heads, Duplicate Token Heads and S-inhibition heads. Specifically for the S-inhibition head `7.0`, when we ablate the head, the original name movers stop preferring the IO tokens but start to promote both IO and S tokens.

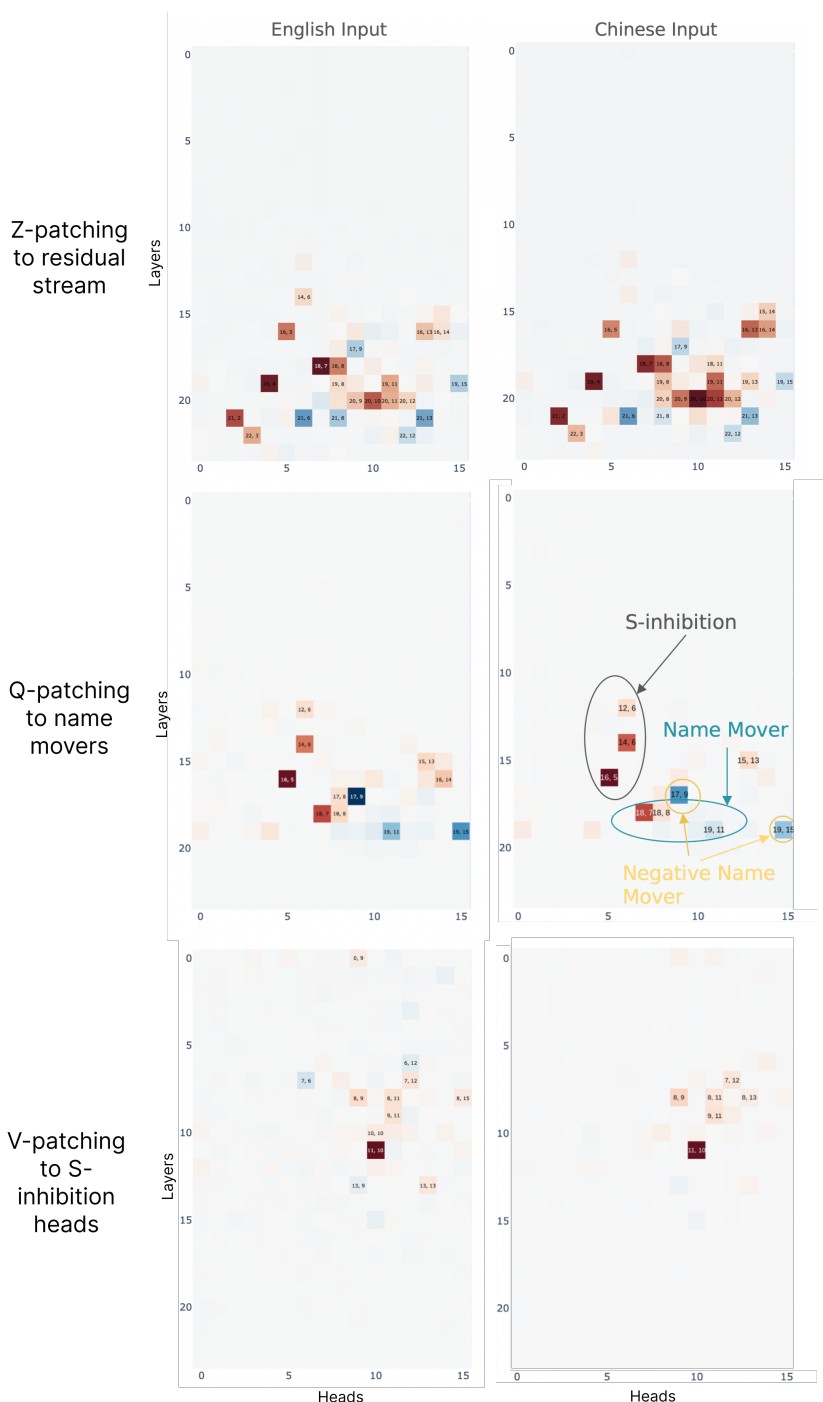

Figure 6: Path Patching steps for extracting IOI circuits on BLOOM-560M. Left: English Input. Right: Chinese Input.

# C  PATCHING DETAILS OF THE PAST TENSE TASK

## C.1  PATH PATCHING

For the Past Tense task, we conduct path patching with the English templates since there are no minimal pairs available for Chinese. As shown in Figure 11 Left, two heads on layer 19 are the most

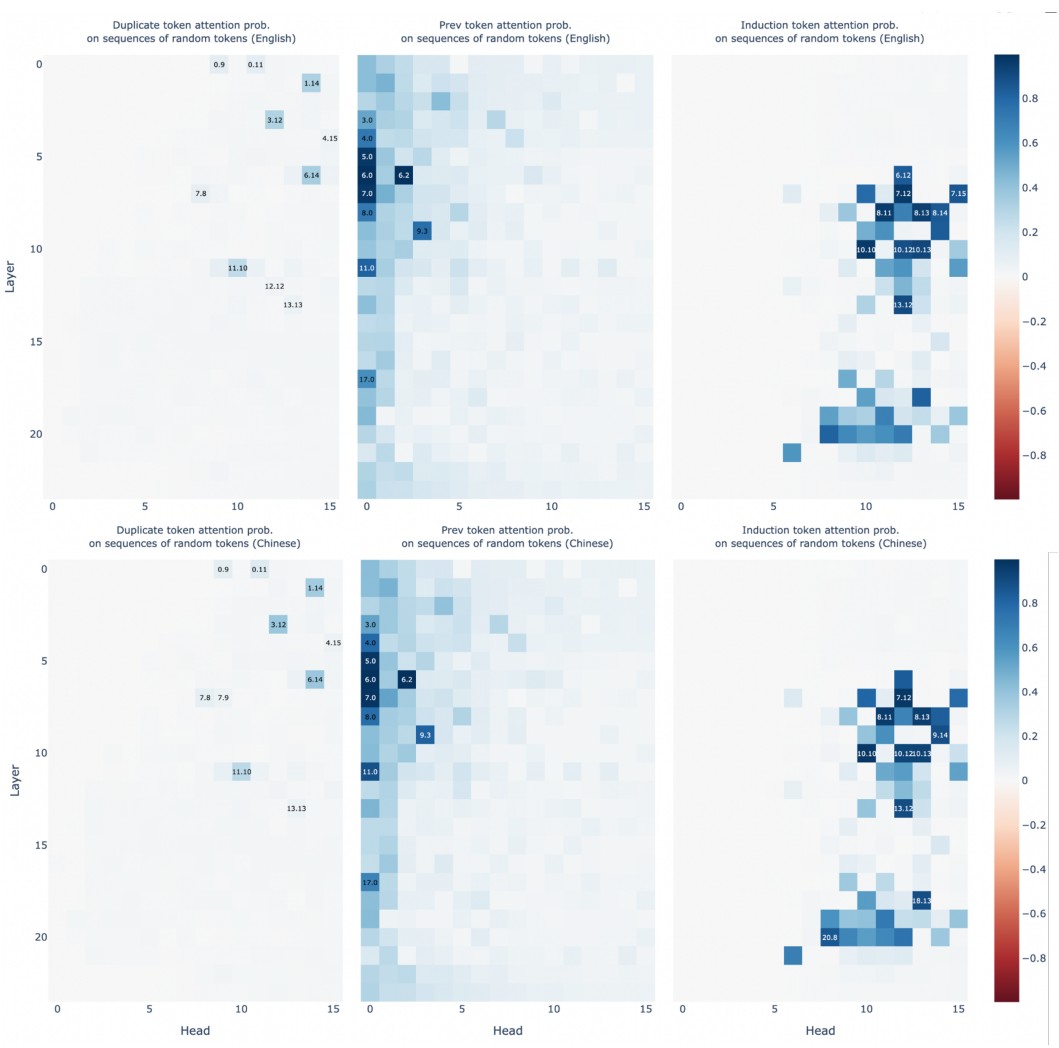

Figure 7: Metrics for Duplicate Token Head, Previous Token Head, and Induction Head on English Inputs (Top) and Chinse Inputs (Bottom).

prominent. We later use head attribution and see that they are past tense heads that pay attention to general past-tense concepts and suppress present-tense verbs (shown in Figure 12). In Figure 11 Right, we project all the head outputs to the vocabulary space and see how much each head writes to the direction of either past tense verbs, present tense verbs in English, or verbs in Chinese. Again we notice that the patterns are similar and a few heads on Layer 21 writes strongly to both past tense and present tense verbs. These heads also show up in the path patching results but do not influence the final logits as strongly as the past tense heads. By examining the top-promoted tokens by these heads, we see that these heads promote orthographically similar or semantically adjacent tokens. These could contain verb forms with various morphological markers as well as synonyms and antonyms (See Figure 12). More detailed discussions can be seen in § 4.5.

## C.2 ABLATION EXPERIMENTS

Figure 12 shows rank changes on the verbs and contains qualitative examples when each type of the function-specific heads are zero-ablated. In general, we notice that past tense heads are specifically activated for English tasks only and therefore the rank of past tense verbs moves forward when these heads are ablated. However, the prediction does not get impacted when we ablate these heads for the Chinese task. Copy heads, on the other hand, are important for both English and Chinese tasks.

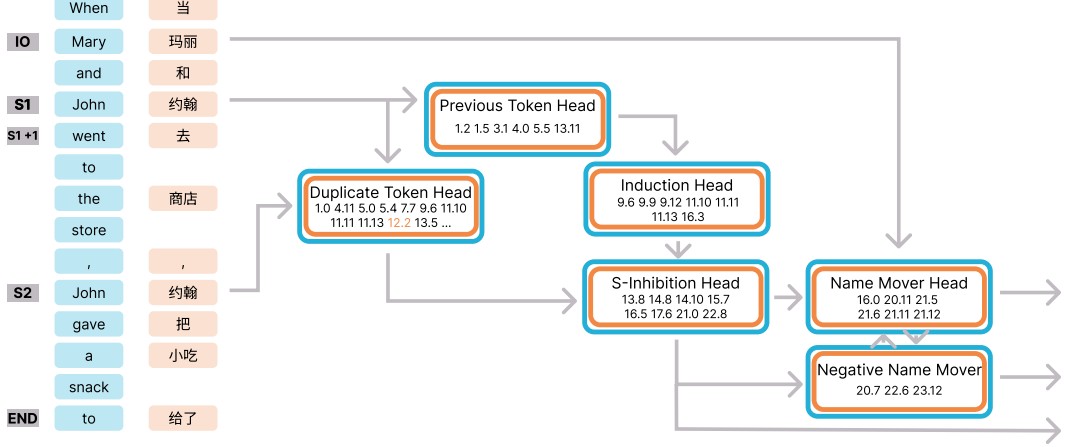

Figure 8: Illustration of IOI circuits of English and Chinese on Qwen2-0.5B-instruct model. Blue-framed rectangles or texts are the components used in English IOI and orange-framed ones are for Chinese IOI.

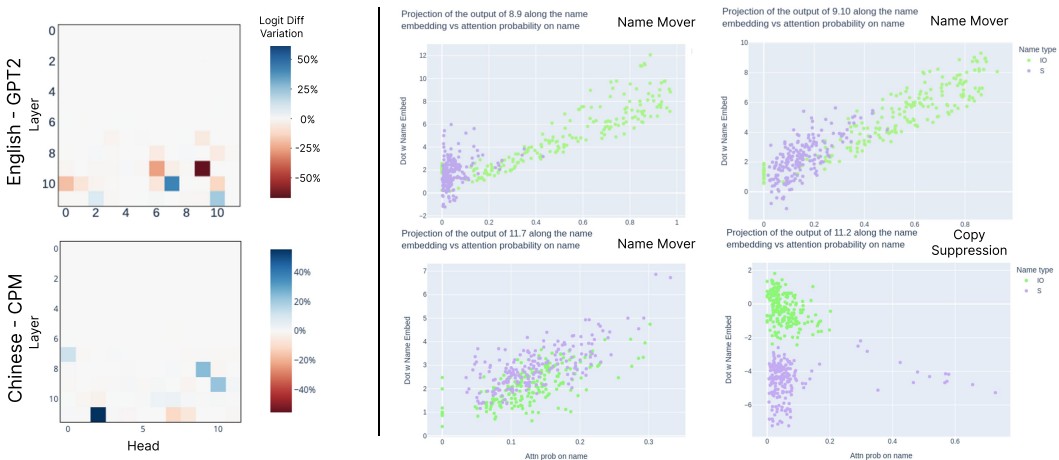

Figure 9: Left: Path patching comparison between GPT2 on English task and CPM on Chinese task. Right: Attention probability vs. head projection output into the IO and S space.

Past-tense, present-tense verbs in English and Chinese verbs are all demoted significantly when copy heads are ablated, indicating these heads are shared components implementing the same functions for both English and Chinese circuits.

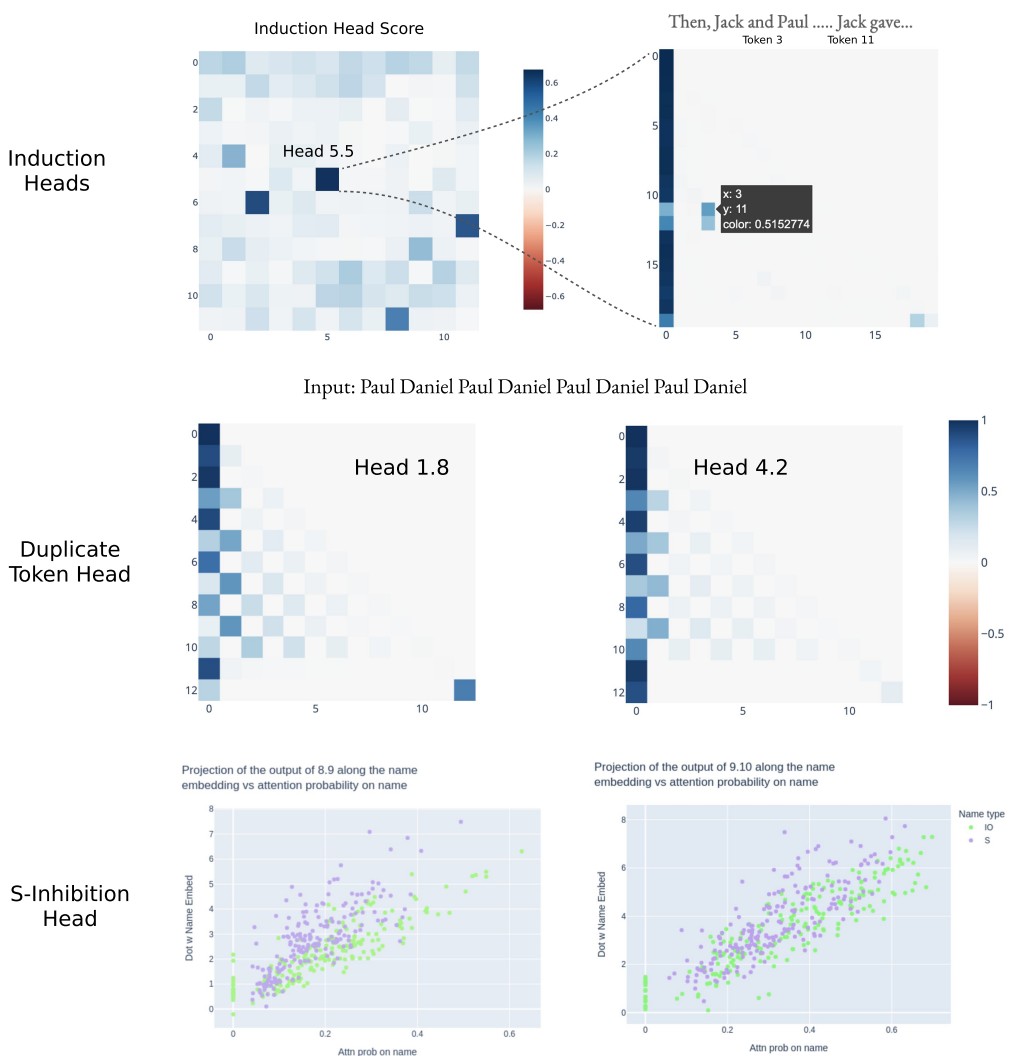

Figure 10: Top: Induction head scores on random tokens and its attention pattern. Middle: Duplicate token attention patterns on ABAB random strings. Bottom: S-inhibition head ablation effects on name movers.

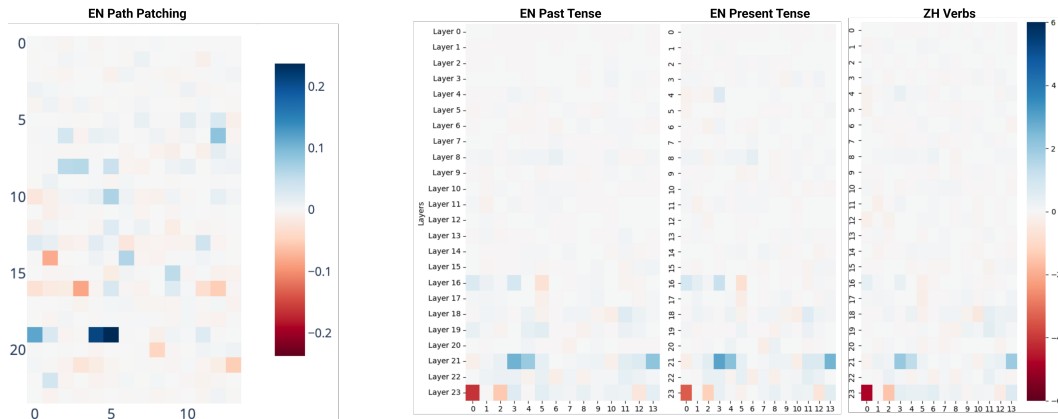

Figure 11: Left: Path Patching on English Past Tense Task. Right: Verb Promotion Heatmaps for English and Chinese Past Tense Task.

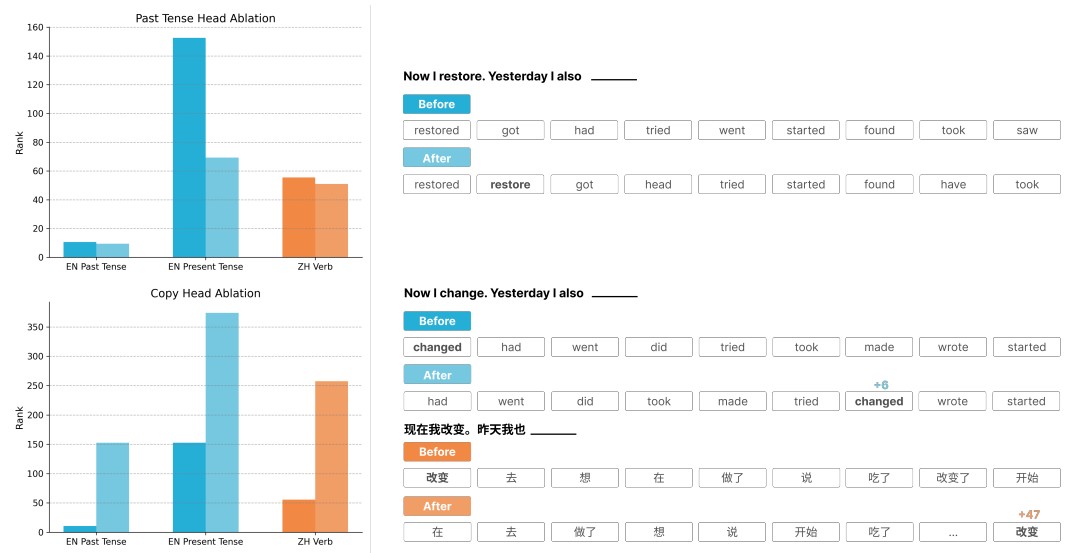

Figure 12: Effects of Past Tense Heads and Copy Head Ablation.

