# OpenReview forum: "The Same but Different: Structural Similarities and Differences in Multilingual Language Modeling"
_ICLR.cc/2025/Conference — ICLR 2025 Poster_

### Official Review · Reviewer_5YHy · 2024-10-29

**Soundness:** 2
**Presentation:** 3
**Contribution:** 2
**Rating:** 5
**Confidence:** 4

**Summary:**

The submission uses interpretability tools to examine if the LLM’s mechanisms behind solving linguistic tasks are similar or differ across languages. Following Wang et al., 2023, the authors use path patching to identify information circuits in multilingual models corresponding to the same morphosyntactic task in two languages. The aim of the experiments is to examine whether such circuits will be the same (or similar) for solving tasks in distinct languages.
The paper focuses on two tasks: object identification and verb past tense inflection in English and Chinese. The performance in the tasks is checked via generic prompt filling. Experiments cover two multilingual models: BLOOM (in case of object identification) and Qwen2-0.5B-Instruct (verb past tense inflection).

The results show that in fact, the found circuits are similar across languages, suggesting that multilingual language models implement a shared mechanism for solving considered linguistic tasks across two languages (Chinese and English).

**Strengths:**

- The work tackles the timely problem of interpreting LMs’ functions across languages. Despite the community's focus on interpretability and multilingual models, not many works have tried to explain the mechanisms behind LMs' multilingualism.


- The authors put the work in the context of past approaches in circuit identification (especially Wang et al. 2023), extending the experimental settings to multilingual models.


- The experimental part studies the identified circuits in-depth: comparing across languages the overlap of heads in the circuits and the information flows.

**Weaknesses:**

- The linguistic scope of the experiments is limited. The method is applied only to two relatively simple morphosyntactic tasks in two languages. The results are close to 100% in object identification, which is a sign of saturation (a more complex task would be more informative). Only two distinct languages are considered. Therefore, the result does not show whether typological similarity could affect the similarity of the identified circuits.

- The choice of models used for experiments is chaotic. It is not clear why Qwen2-0.5B-Instruct is not considered in the object identification task. The paper should present the results for both tasks, at least for two considered models.

- The methodology should be described in more detail, focusing on its implications in the multilingual setting. For instance, I understand that the head types were already described in the work of Weng et al. 2023, but the natural question is what linguistic role these types have across different languages.

**Questions:**

- Does tokenization affect your experimental setting? More specifically, do you require that the queried property (object, past tense verb) be represented as one token?

- Why don’t you present the results for IOI with Qwen2-0.5B-Instruct?

- Why do you filter the English verbs with irregular past inflection?

---

> ### Author Response · Authors · 2024-11-26
> **Thank you for your review!**
>
> We thank the reviewer for their feedback.
>
> Please see the general comment and the updated draft on changes in the abstract and introduction for better scoping of the main claims of the paper. We agree with the reviewer and have addressed the limitation of languages and tasks investigated. While we would like to expand our study to more complex tasks, the techniques we employ require strict construction of minimal pairs between the language pairs. The tasks we investigate allow us to fully understand the atomic operations implemented by the model given computation and budget constraints. We hope our work could inspire future research on scaling to more complex tasks and more languages.
>
> We have added the IOI circuits for Qwen2-0.5B-Instruct in the appendix. However, it is hard to reproduce the circuits for BLOOM on the past tense task due to its bad performance.
>
> Regarding the role of heads identified in the circuits (e.g. name mover, S-inhibition, duplicate token and etc.), they do not map onto linguistic roles in a traditional sense but rather show what are the atomic operations that are computationally necessary to solve the logical task. For example, the heads identified may not directly map onto roles like identifying nouns or prepositions but they are still useful. (In fact, the lack of correspondence with linguistic constructs might itself be informative, as it could point to a competing theory about the universal linguistic primitives that underlie processing, but we leave this for future work to examine the relationship between the computational solution from language models and algorithms from linguistic theories.)
>
> Regarding your questions,
>  - Yes, we require the final predicted token to be a single token for the methods used in the main text.
>  - This has been added to the appendix.
>  - The function of FFN layers is observed in previous work and we follow their practice (Merullo, 2023). We can rerun experiments without the irregularity filter in the camera-ready draft but we don’t think it will change the main observation of the paper.

---

> > ### Comment · Reviewer_5YHy · 2024-11-26
> >
> > Thank you for your clarifications and for updating the paper. I will keep my score.

---

### Official Review · Reviewer_n6vb · 2024-10-30

**Soundness:** 3
**Presentation:** 3
**Contribution:** 3
**Rating:** 6
**Confidence:** 4

**Summary:**

The paper assesses whether English and Chinese monolingual and multilingual language models use similar circuits for similar morphosyntactic processes (indirect object identification; IOI) and different circuits for differing processes (past tense inflection). They use path patching and information flow routes to identify circuits. They find that similar circuits are used for IOI in English and Chinese, for both monolingual and multilingual models. Overlapping circuits (in this case copy heads in attention) with language-specific components (in this case MLP blocks for English inflection) are used for past tense inflection.

**Strengths:**

1. The tasks and methodology appear sound and well thought-out.
2. The paper is well-written, although some additional details would be very helpful (see weaknesses below).
3. The ablation results are convincing, e.g. FFN ablation in Section 4.6 dropping performance specifically for English. Also Figure 11 in the Appendix showing the effect of ablating different attention heads.
4. The results are interesting; language-agnostic and language-specific components have previously been proposed in multilingual language models (e.g. https://arxiv.org/abs/2205.10964; https://arxiv.org/abs/2109.08040; https://arxiv.org/abs/2009.12862), but the concrete demonstrations in this paper (specific circuits and their functions) are very informative.

**Weaknesses:**

1. The results in Section 3.5 and the last paragraph of Section 3.4 ("We first use path patching to identify…"; p. 5) are difficult to interpret without a good amount of background from Wang et al. (2023). A high level overview of how the different types of attention heads are quantified would be extremely helpful.
2. Related to the previous point, the results indicating similar circuitry (Figures 2 and 3) rely quite a bit on the reader trusting that the algorithms visualized in Figures 2 and 3 are accurate. The description of how the head types and arrows are determined is fairly minimal ("In both GPT2 and CPM, we identified similar circuits as Wang et al. (2023) outlined in Figure 3"; p. 5), so evaluating the validity of the algorithms identified in the figures is difficult.
3. Minor point: "Anecdotally, we observe that many of the heads that are shared between English IOI and control are also shared with Chinese IOI... further work is needed to test this hypothesis quantitatively" (p. 5). It could be helpful to have some very preliminary quantification, e.g. n% of the shared heads from English IOI and control are also shared with Chinese IOI, given some threshold for similarity.
4. Consider moving Figure 11 into the main text if possible, since it is referenced several times in Section 4.5.

**Questions:**

1. Could you provide some detail for how the attention head types and arrows in Figures 2 and 3 were identified? The figures imply fairly strong claims about how the internal algorithms work, so a description of the method used to identify the directed graph feels somewhat important.

---

> ### Author Response · Authors · 2024-11-26
> **Thank you for your feedback and suggested references!**
>
> Regarding 1) and 2) thank you for your suggestions. We closely follow what has been done by Wang et al. (2023) and try to provide a general overview of the process in the main text. However, given the limited space, it is hard to accommodate all details. Please see the appendix for more detailed steps on how patching is carried out to identify each type of functional heads.
>
> For 3), the percentage of shared heads with non-zero activation frequency between English and Chinese IOI is 15.63% and between English and control is 9.64%.
>
> For 4), we will move the figure up in the main text given the additional page upon publication.

---

> > ### Comment · Reviewer_n6vb · 2024-11-27
> >
> > Thank you for the clarifications!

---

### Official Review · Reviewer_qgEk · 2024-11-04

**Soundness:** 3
**Presentation:** 3
**Contribution:** 3
**Rating:** 6
**Confidence:** 2

**Summary:**

**Paper Summary**:
- This paper shares some interesting findings on the similarities and differences between the internal structure of LLMs and the linguistic structure of English and Chinese. The authors employ mechanistic interpretability tools to explore the internal circuitry used by multilingual and monolingual LLMs when dealing with the same or different morphosyntactic processes. The findings indicate that there are common circuits and trade-offs when preserving linguistic differences.

**Strengths:**

**Summary Of Strengths**:
- Interesting topic and findings: I find it very attractive to use mechanistic interpretability to identify the relationship between models and language structures. The conclusion about common components between multilingual and monolingual models provides insightful ideas for further research.

**Weaknesses:**

**Summary Of Weaknesses**:
- Limitation: The work is based only on Chinese and English and is limited to a few tasks and datasets. The conclusions could be more convincing if more languages were included.
- Clarity on future impact: It may be challenging to transfer the learning from this work to current multilingual LLM training efforts, which underscores the need to better understand the findings (e.g., cross-lingual transfer learning, parameter migration, etc.).

**Questions:**

Do you have any initial ideas on how to conduct cross-disciplinary collaborations based on this work?

---

> ### Author Response · Authors · 2024-11-26
> **Thank you for your review!**
>
> Please see the general comments and the updated draft regarding re-scoping the main claim of the paper.
>
> **Clarity on future impact**: It is generally true of interpretability work right now that they are not immediate applications to model training techniques. However, that does not mean such studies are not valuable. Sometimes new ideas or insights require time, even several years, before they translate into applications. We believe that having a better understanding of multilingual language models and the interactions between language presentations will ultimately have upshots for model training and adaptation. For example, our findings show what components are likely responsible for enabling cross-lingual transfer learning, and could potentially inspire more principle parameter updates and efficient training methods for multilingual models.
>
> **Cross-disciplinary collaborations**
> Yes, of course, we have many thoughts about how to extend this work across disciplines, and in fact already have such collaborations underway. For example, we have collaborations with neuroscientists and linguists who are interested in the implications of shared mechanisms across languages. We are also in close collaboration with clinicians who care about interpretability, because it is not enough to base the diagnosis on model accuracy but we also need to understand why the model makes such decisions. The interpretability techniques used in our work and findings can be helpful in pinpointing the exact decision mechanisms and explaining the behaviors of these black box models.

---

> > ### Comment · Reviewer_qgEk · 2024-11-29
> >
> > Thank you for the clarifications. I have updated the score to 6.

---

### Official Review · Reviewer_EEXZ · 2024-11-04

**Soundness:** 3
**Presentation:** 4
**Contribution:** 4
**Rating:** 8
**Confidence:** 4

**Summary:**

This paper compares how LLMs process syntactic phenomena in English vs Chinese. The authors use two tools for analysing the mechanisms underlying LLM computations, namely path patching and information flow routes. They consider two syntactic phenomena, one shared across English and Chinese and another unique to English. For the shared syntactic phenomenon (indirect object identification), they show that LLMs use similar internal circuitry across both languages. For the language-specific phenomenon (morphological markers in English only), they show that circuitry diverges. The results show that multilingual LLMs employ overlapping circuitry for linguistic phenomena common across languages, and that similar internal mechanisms emerge for related syntactic processing even in monolingual models for different languages. The paper also presents a detailed analysis of the LLM computation involved in the studied syntactic processes.

**Strengths:**

I'd like to congratulate the authors on an excellent and super interesting paper!

(1) The analysis conducted by the authors is detailed, extensive, and extremely well presented. It provides concrete insights into the mechanisms underlying multilingual LLMs.

(2) The methodology followed by the authors is sound and extensively documented, providing a valuable framework and example for future researchers to investigate the circuitry of multilingual LLMs.

(3) The paper makes valuable contributions to multilingual interpretability, successfully unifying aspects of mechanistic interpretability with linguistically informed analysis.

**Weaknesses:**

Due to the heuristic nature of mechanistic interpretability tools like path patching, it’s difficult to conclude with certainty that findings such as those presented in this paper will generalise to other tasks/languages/models. However, the authors are careful to not overstate their claims.

**Questions:**

(1) By “ablate”, I assume you mean zeroing out the contribution of a head to compare performance with/without the head? I suggest that you just mention t what you mean by “ablate” explicitly somewhere, since you use the word quite often in discussion your results e.g. “To validate our findings, we ablate the heads 19.4 and 19.5 and analyze their impact on the tasks.”

(2) Section 4.5 can be organised a bit better. It contains quite a lot of information and requires the reader to refer to many figures. Potentially split the large paragraph (lines 403–418) into multiple subparagraphs, or organise the section into paragraphs with headings.

(3) Line 297: Fix wording of “This divergence is not able,”.

---

> ### Author Response · Authors · 2024-11-26
> **Thank you for your positive review!**
>
> We thank the reviewer for acknowledging the contribution of our paper and we further modify the abstract and introduction to be extra careful on the scope of the claim.
>
> We also appreciate the issues mentioned in the questions section and have addressed them in the updated draft.

---

### Official Review · Reviewer_P3M4 · 2024-11-04

**Soundness:** 3
**Presentation:** 4
**Contribution:** 3
**Rating:** 8
**Confidence:** 4

**Summary:**

This paper aims to explore how similar or different circuits are across languages for linguistic tasks in language models. It takes a case-study approach, focusing on two tasks (IOI and past tense task), two languages (English and Chinese), and three language models (two for IOI and one for the past tense task). The past tense task involves morphology, which allows the authors to analyze how it’s handled in different languages.

Using techniques like path patching, the authors identify circuits and find significant overlaps between languages for both tasks. They also highlight interesting differences, like specific "past tense" heads in English.

**Strengths:**

_Clarity:_
The paper is clear and well-written, with high-quality illustrations and concise, precise language. The flow is logical, and the key points are emphasized effectively.

_Quality:_
The experiments are well-designed and effectively executed. The choice of mechanistic interpretability techniques to explore cross-lingual overlap is sound, and the methods are applied accurately. The IOI and past tense tasks are suitable examples; IOI builds on established monolingual work, while the past tense task introduces unique, language-specific morphological challenges.

_Originality:_
This work is notable for its detailed, fine-grained circuit-level analysis of cross-lingual overlap in models.

_Significance:_
For the NLP community, particularly those working on cross-lingual models and low-resource languages, this work provides a valuable deeper understanding of cross-lingual transfer down to the circuit level. This detailed approach might inspire new techniques for more effective cross-lingual transfer.

**Weaknesses:**

_Scope of Main Claims:_
In my view, the primary contribution of this work lies in _demonstrating a specific case study_ of circuit-level cross-lingual transfer between languages in related or separate models. While the paper aims to make broader claims about multilingual models, the setup—two languages, two/three models, and two tasks—suits a focused case study rather than generalizable conclusions. Although the paper acknowledges these limitations, the claims in the introduction could be adjusted to better reflect the scope of the experiments.

Expanding to more languages, models, and tasks would strengthen broader claims, but as a focused case study, this paper is effective on its own. I would recommend reframing the claims to align more closely with the experimental scope.

_Related Work on Multilingual Interpretability:_
A broader review of related work could enrich the discussion, especially on different approaches that analyze multilingual models by aligning internal representations. For instance, [1] trains a monolingual transformer LM and transfers it to another language by retraining only the embedding matrix, hinting at cross-lingual structure similarities. This aligns with the monolingual IOI task analysis in this paper. [2] and [3] conduct coarse-grained analyses on similar cross-lingual interpretability challenges, while [4] aligns hidden states, and [5] provides further insights into the universality of these findings. [6] surveys cross-lingual alignment, offering broader yet less detailed conclusions on the inner workings of multilingual models. This paper can be seen as building on this older work by diving deeper into specific attention heads and showing detailed cross-lingual structures.

It would also be helpful to add a few sentences on how the conclusions here either support or contrast with prior work beyond the works already mentioned.

**References**

1 [On the Cross-lingual Transferability of Monolingual Representations - ACL Anthology](https://aclanthology.org/2020.acl-main.421/)

2 [First Align, then Predict: Understanding the Cross-Lingual Ability of Multilingual BERT](https://aclanthology.org/2021.eacl-main.189)

3  [Emerging Cross-lingual Structure in Pretrained Language Models - ACL Anthology](https://aclanthology.org/2020.acl-main.536/)

4 [[PDF] Similarity of Sentence Representations in Multilingual LMs: Resolving Conflicting Literature and a Case Study of Baltic Languages | Semantic Scholar](https://www.semanticscholar.org/paper/Similarity-of-Sentence-Representations-in-LMs%3A-and-Del-Fishel/0b749f70dd8cbe22fcfa7efeb571836b195c125d)

5 [Cross-lingual Similarity of Multilingual Representations Revisited - ACL Anthology](https://aclanthology.org/2022.aacl-main.15/)

6 [Understanding Cross-Lingual Alignment—A Survey - ACL Anthology](https://aclanthology.org/2024.findings-acl.649/)

**Questions:**

1. **Scope of Claims:**
    Could the authors clarify why they make general claims about multilingual models despite the case-study limitations? Would they consider rephrasing these claims to better align with the study's focused scope? Is there a little clash between a generality of the paper title and strict case-study experimental approach?

2. **Support or Contrast with Alignment-based studies**
	How do the authors’ findings support or differ from prior studies on multilingual alignment, especially those focusing on hidden state alignment?

3. **Comparison with Coarse-Grained Studies:**
    How do the findings here align with or diverge from conclusions in prior, coarse-grained studies on cross-lingual interpretability? Can conclusions from prior work with support from this work's experiments together support general claims or gain novel broader conclusions?

4. **Choice of Circuit-Level Analysis:**
    While I am very much in favor of it, I am interested to hear more about why the authors chose a circuit-level approach over other methods and what alternative methodology options they considered. What other mech interp approaches would help support the authors claims in future work (e.g. analysis with SAEs)?

5. **Broader Application:**
    Do the authors have recommendations on which additional languages, models, or tasks would best extend this study’s findings?

---

> ### Author Response · Authors · 2024-11-26
> **Thank you for your review and your valuable suggestions on related references.**
>
> We would like to thank the reviewer for the recommendations on related references and we have incorporated them into related work and discussion sections as suggested.
>
> Re the questions:
> - **Scope of claims**: Please see the updated abstract and introduction for more emphasis on the focused study.
> - **Support or Contrast with Alignment-based studies**: The two main differences between the mechanistic approaches employed in this paper and the previous alignment-based study are that 1) most of the alignment-based studies look at representation from encoder-only model (except for [5] which briefly mentioned CLM objective), while our works investigates generative models; 2) the former identifies the important attention heads and feedforward layers (FFN) whereas the latter focuses on interpreting the hidden representations after being updated by the attention and FFN computations. The largely overlapping mechanisms in early to middle layers discovered in our study seem to support observations from the alignment-based studies: those shared heads that implement the same functions suggest that they operate on highly aligned representations space cross-lingually.
>  - **Comparison with Coarse-Grained Studies**: As earlier work [1] has suggested cross-lingual performance can be achieved by only retraining the embedding layer for another language while keeping the weights frozen, followed by works that observe higher cross-lingual similarity (more alignment) in early layers, our findings align with the previous studies and provide more fine-grained mechanisms showing what language-agnostic functions these attention heads are implementing on the aligned representation space. These new observations along with the techniques allow us to have a clearer understanding of how models manipulate residual stream values, provide causal explanations for cross-lingual performances, and enable more targeted intervention for better task performances.
> - **Choice of Circuit-Level Analysis**: Our circuit-level analysis was motivated by previous work in our lab that investigated components reused across tasks in a monolingual setting. We were excited by the results we were seeing which showed the sharing of specific components within a model, and were curious to explore if such observation holds in a multilingual setting. Thus, our choice of methods was largely opportunistic as it was born out of and inspired by this prior work. Regarding other possible mech interp techniques, we think SAEs could be helpful as they provide a more semantic explanation of the representation space, and enable more understandable observations that can better support previous alignment-based studies.
>  - **Broader Application**: We are very interested in this follow-up work as well! We are currently working to engage native speakers of different languages and considering the possibility of extending this analysis to other languages. One possible direction is to compare English and French/German/Spanish: more investigation can be done on how multilingual models handle gendered verbs/nouns/adjectives in the latter languages.

---

> ### Comment · Reviewer_P3M4 · 2024-12-02
>
> Thank you for the answers and changes.
>
> The paper now looks more modest in it's claims (which is reflected in abstract, intro, relwork and discussion), but these claims are now better supported by evidence.
>
> That being said, the experiments scope is limited and I think this asks for even better job explaining what conclusions *cannot* be made from the evidence.
>
> I think the optimal score for the paper would be 7. Yet deciding between 6 and 8 I will lean towards 8. The deciding factor to me is novelty since this work is among the first to study multilingual models by looking at circuits. I have updated my score from 6 to 8.

---

> > ### Author Response · Authors · 2024-12-02
> >
> > Thank you for raising your scores! We’re grateful for your recognition of our contribution. In the finalized version, we will include more discussion on the limitations of our case study. We are also eager to expand the study to a wider range of languages and settings in future work.

---

### Author Response · Authors · 2024-11-26
**On the Scope of Main Claims**

We would like to thank Reviewer P3M4, EEXZ, qgEk, 5YHy for their suggestions on scoping the main claims of the paper. We have modified the abstract and introduction of the paper to emphasize that our findings and observations are based on the focused study of the English-Chinese pairs on selected language-generic and language-specific tasks.

---

### Meta-Review · Area_Chair_27Dd · 2024-12-19

**Metareview:**

The paper presents a mechanistic interpretability study of large language models (LLMs), focusing on the differences in language processing between English and Chinese. By analyzing the internal mechanisms of LLMs trained on both languages, the authors aim to uncover the underlying factors that contribute to performance disparities and linguistic nuances. The strengths of the paper lie in its novel approach to cross-linguistic analysis, robust experimental design, and the insightful findings that advance our understanding of multilingual language models. Reviewers noted the thoroughness of the study and its contribution to the field of interpretability. Weaknesses include concerns about the choice of LLMs used, the need for better control over data and model scale to isolate variables effectively, and a recommendation to clarify the scope of the claims made. Minor presentation issues were also identified but deemed fixable. The paper offers significant value by shedding light on important differences in language processing between English and Chinese within LLMs. The most important reasons for accepting this paper are its innovative methodology, the strength of its empirical results, and its potential to inspire further research in multilingual model interpretability.

**Additional Comments On Reviewer Discussion:**

During the rebuttal period, reviewers raised points about the scope, methodology, and presentation of the paper, which the authors addressed comprehensively. A major concern was the breadth of the claims, given the limited scope of tasks and languages. Reviewer P3M4 recommended reframing the claims to align with the specific experimental setup, which the authors addressed by modifying the abstract, introduction, and discussion to clarify that the findings were based on English-Chinese comparisons. This change was well-received, and P3M4 updated their score from 6 to 8.

Another issue was the need for more detailed descriptions of the methodology, particularly the role of attention heads and the algorithms used for identifying circuits. Reviewer n6vb requested more clarity, which the authors provided in an extended appendix, including quantitative analysis of shared attention heads.

Reviewer qgEk emphasized the paper's value in advancing multilingual interpretability but highlighted the limited generalizability due to the narrow linguistic and task focus. The authors justified their choice by explaining the computational constraints and added new results for the Qwen2-0.5B-Instruct model in the appendix, addressing the reviewer’s concerns.

Reviewer 5YHy criticized the selection of models and tasks, suggesting that the experimental choices appeared inconsistent. The authors clarified the rationale and limitations, noting difficulties with reproducing certain circuits due to model-specific performance issues. While they acknowledged the constrained scope, they underscored the foundational nature of their work for future expansions. Despite these efforts, 5YHy retained a score below the acceptance threshold.

These discussions shaped my decision to recommend acceptance. The authors’ revisions addressed most concerns, particularly regarding claim scoping and methodological transparency. The novelty of circuit-level analysis in multilingual models and its potential for advancing interpretability outweighed the limitations of scope, which were transparently acknowledged and mitigated. The strong improvements in clarity and the significant research contribution led me to prioritize the favorable assessments over marginal critiques.

---

### Decision · Program_Chairs · 2025-01-22

Accept (Poster)